# Remote-Controlled Drone System through Eye Movements of Patients Who Need Long-Term Care: An Intermediary’s Role

**DOI:** 10.3390/healthcare10050827

**Published:** 2022-04-30

**Authors:** Feni Betriana, Ryuichi Tanioka, Atsunori Kogawa, Riku Suzuki, Yuki Seki, Kyoko Osaka, Yueren Zhao, Yoshihiro Kai, Tetsuya Tanioka, Rozzano Locsin

**Affiliations:** 1Graduate School of Health Sciences, Tokushima University, Tokushima 770-8503, Japan; fenibetriana@gmail.com (F.B.); taniokaryuichi@gmail.com (R.T.); 2Department of Mechanical Engineering, Tokai University, Kanagawa 259-1292, Japan; ironuta.kk@gmail.com (A.K.); riku.suzuki0616@gmail.com (R.S.); seki.yuxi@gmail.com (Y.S.); kai@keyaki.cc.u-tokai.ac.jp (Y.K.); 3Department of Clinical Nursing, Kochi Medical School, Kochi University, Nankoku 783-8505, Japan; 4Department of Psychiatry, Fujita Health University, Aichi 470-1192, Japan; taipeeyen@gmail.com; 5Institute of Biomedical Sciences, Graduate School, Tokushima University, Tokushima 770-8509, Japan; tanioka.tetsuya@tokushima-u.ac.jp (T.T.); locsin@tokushima-u.ac.jp (R.L.)

**Keywords:** drone, control system, clinical experiment, long-term hospitalization, robot, eye-tracking device, quality of life, intermediary role, nurse

## Abstract

The use of a remote-controlled drone system (RDS) by eye movements was studied to assist patients in psychiatric long-term care (PLTC) to allow them to view the environment outside the hospital, hoping that this will bring them some enjoyment. However, successfully applying this system requires human intermediaries in facilitating the interactions between patients and RDS operators. The aim of the study was to describe the role of nurses as intermediaries in the application of an RDS through eye movements of patients PLTC. This study employed the Intentional Observational Clinical Research Design. Data collection was performed in November 2021 at a psychiatric hospital with selected patients in PLTC. Seventeen patients took part in the indoor experiment, whereas 23 patients took part in the outdoor experiment. Fifteen of the 23 patients in the outdoor experiment were the same patients who took part in the indoor experiment. Most of the patients in the indoor and outdoor test arenas could successfully, delightfully, and safely fly the drone. This study demonstrated that RDS using just eye movements could increase the quality of life in older patients with psychiatric problems in PLTC. For the successful use of this drone system, nurse intermediaries assumed critically significant roles.

## 1. Introduction

The world is experiencing a super-aged society, in which the population of older people continues to increase annually. In 2019, there were 703 million people aged 65 years and older worldwide, and this number is estimated to double to 1.5 billion in 2050 [1]. Unfortunately, the increasing older population does not always have a good quality of health and is often dependent on others for assistance.

As people get older and become frail, many of them require long-term care [2]. Today, a person turning 65 has an almost 70% chance of requiring some type of long-term services and support. Consequently, one-third of older adults who are aged 65 years old today may need long-term care support, of which 20% will need it for longer than 5 years [3]. In the US, about 1.5 million older people live in nursing homes, and 1 million live in assisted living facilities [4]. Meanwhile, in Japan, the number of people using long-term care services on a monthly basis reached approximately 5.6 million [5]. Though long-term care is necessary for older people, it can affect their psychological condition, because they have less contact with the outside world. Patients with long-term hospitalization, especially those who are bed-ridden, might not be able to see outside the hospital, which makes them quite stressed and leads to a deterioration in their quality of life (QOL).

Various information communication technologies and robot systems have been developed to help patients, particularly older people, to eliminate loneliness, and to maintain and improve their QOL [6,7]. These are robot systems that aim to provide healing to a patient by having them interact with the robot. Alternatively, as one method for alleviating the loneliness of patients, a drone system expected enabling patients with mental illness to communicate with people in distant places, while also enjoying the scenery in the distant places in real time, using a camera mounted on the drone. Moreover, unlike robots that move using wheels and legs, drones can fly over steps and stairs; therefore, it is thought that the stress of patients with mental illness will be reduced by operating drones. In addition, for patients with mental illness (hereinafter ‘patients’) without opportunities to go out of the hospital because of undergoing long-term medical treatment, it is considered that interacting with people through drones may lead to healing.

Drones have been used in healthcare services, such as transporting blood, vaccines, and other supplies to remote areas [8]. In a clinical trial by Schierbeck et al. [9], automated external defibrillator-equipped drones arrived before ambulances for dealing with out-of-hospital cardiac arrests. Aggarwal et al. [10] conducted a study using a nano-drone with a camera to monitor patients in the intensive care unit. Meanwhile, Kai et al. [11] evaluated a remote-controlled drone system using an eye-tracking device through the internet to improve the quality of life for patients in bedridden conditions.

However, controlling the drone system using eye tracking might not be easy for older patients who have been bedridden for a long time. Not being familiar with sophisticated advanced technology could also be a challenge for older patients to operate the drone. Hence, the role of intermediaries to assist patients and to facilitate the application of this technology is necessary. Intermediaries, who are generally nurses, play an important role in advocating for patients, and in mediating and associating the patients in the transactive relationship involving intelligent machines, such as healthcare robots [12]. In the use of communication robots in a community, intermediary roles include understanding the performance and functions of robots, identifying issues and concerns of ethics, morality, security, and safety in using robots, and ensuring physical and mental safety of the community residents during their interactions with robots [13]. Though the roles of intermediaries in using robots have been discussed in several studies [13,14,15], the roles of intermediaries in the drone system for patients with psychiatric long-term hospitalization have not yet been investigated.

The aim of the study was to describe the role of nurses as intermediaries in the application of a remote-controlled drone system through the eye movements of patients in long-term care.

## 2. Materials and Methods

### 2.1. Design

This study employed the Intentional Observational Clinical Research Design (IOCRD) [16]. IOCRD is a composite of observational research approaches that are aimed to generate data from a simultaneous procedure. IOCRD intended to address phenomena involving the interaction between patients, intermediaries, and technologies with artificial intelligence [16]. In this study, the researchers applied the intentional observation aspect of IOCRD in order to observe the intermediary’s roles, which enables patients with long-term hospitalization to remotely control the drone using only their eyes.

### 2.2. Overview of the Operating System

This section describes the drone system and the operation method used in this study.

#### 2.2.1. System

Figure 1 shows the remote-controlled drone system. This system can be controlled automatically by combining independently developed systems (equipment and software) and commercially available systems (equipment and software). This system includes an operation screen, an eye-tracking device (Tobii Eye Tracker 4C, Tobii), computer A, computer B, Internet, a drone (Mavic Mini, DJI), a controller, a smartphone (iPhone XR, Apple), and the web-conferencing service Zoom.

The operation method of the drone is as follows: The eye-tracking device attached to the operation screen detects the patient’s gaze position in the operation screen; the data of this gaze position are sent to computer B at a distant location via computer A and the Internet. The drone is operated via computer B, and the controller is based on the gaze position data. The drone’s camera image is displayed on the operation screen via the controller, the smartphone, Zoom, and computer A. Then, the patient can see the scenery of a distant place on the operation screen.

#### 2.2.2. Operation Screen/Operation Method

An operation screen was installed, so that patients who need long-term care confined to their beds can easily control the drone system with this operation screen. The operation screen of this system and the coordinates around the drone in flight are shown in Figure 2. This operation screen is divided into six areas. Area 1 is a neutral circle in the center of the operation screen and excludes area 2. Area 2 is a forward circle below the center of the operation screen and inside area 1. Line A and line B are the tangent lines to the neutral circle in the Z direction. Area 3 is the area on the right side of line A, and area 4 is the area on the left side of line B. Area 5 is the area above area 1, and area 6 is the area below area 1.

The operation method of the drone is as follows: (1) looking at Area 1, the drone is hovering; (2) looking at Area 2, the drone moves forward; (3) looking at Area 3, the drone rotates to the right; (4) looking at Area 4, the drone rotates to the left; (5) looking at Area 5, the drone rises; and (6) looking at Area 6, the drone descends.

On the operation screen, “right,” “left,” “up,” “down,” “forward,” and “stop” are written in Japanese characters so that the patient does not forget the operation method.

### 2.3. Settings and Participants

This study involved several settings: (1) a laboratory in Tokai University from where the systems of the drones were controlled, (2) a psychiatric hospital ward in Western Japan from where the patients moved the drone with their eye tracking, (3) a large gymnasium in Tokai University where the drone was placed for an indoor experiment, and (4) a rooftop in Tokai University where the drone was placed for an outdoor experiment.

Furthermore, this study involved several participants: (1) two operators from the research team who set up and controlled the drone system, (2) nursing staff, with the role as intermediaries who were familiar with and had experience being involved in the previous experiment using advanced technologies in the ward (e.g., experiment with healthcare robot), (3) patients in the ward who moved the drone by their eye tracking, and (4) students with the roles of checking and ensuring the movement of the drone was within the expected area, and assisting to entertain the patients by waving to the drone camera, which was seen by the patients.

The inclusion criteria for patients included: (1) the ability to follow the instructions from the drone operators, and (2) successful calibration of eye movements for eye tracking.

### 2.4. Data Collection Procedure

Data collection was conducted in three phases of the clinical research process: (1) pre-experimental phase, (2) experimental phase, and (3) post-experimental phase. The period of drone operation by each patient was around 5–7 min.

There were two types of experimental phases: indoor and outdoor experiments. In the indoor experiment, the drone was flown inside the large gymnasium of Tokai University, whereas in the outdoor experiment, the drone was flown from the rooftop of a building at Tokai University. With these experimental phases, the participation of patients was directed toward engagement in the manipulation of the experimental device (indoor experiment) and patient manipulation of the device to test for patients’ ability to view the outdoor scenery, such as Mount Fuji, capitalizing on the significance of the study, i.e., for patients who are limited by mobility to view the environment outside of their patient rooms.

#### 2.4.1. Pre-Experimental Phase

In this phase, patients’ eyesight was calibrated to enable the manipulation of the eye-tracking device. The calibration phase was conducted in the same manner for both the indoor and outdoor experiments. The calibration process was conducted in the laboratory of Tokai University (for remote operation) and in the ward of the hospital setting (operator for patients’ eye-tracking calibration).

There were two operators setting the calibration: one remote operator set the drone system at Tokai University (Hiratsuka City, Kanagawa Prefecture), and another operator conducted the calibration for eye tracking with patients in the hospital setting. Both operators were members of the research team.

First, the operators explained the operation method to the subjects. Next, the eye- tracking device was calibrated for each patient. The calibration process involved the operator, intermediaries near the patients, and the patients themselves to move the drone using their eyesight with the support of the intermediaries.

The operator in Tokai University communicated with the operator, intermediaries, and patients in the hospital using a speaker, a microphone, and Zoom application. In addition, next to the patient, computer A was placed and connected to the Internet using mobile Wi-Fi (DISM WiMAX2 + Package Education New Year Edition, DIS mobile (WiMAX)). Figure 3 shows the situation of the operator at Tokai University, and Figure 4 shows the ideal position of the patient in the hospital setting.

#### 2.4.2. Experimental Phase

Patients could start operating the drone once the calibration was successful. In the indoor test, the drone was flown in a large gymnasium at Tokai University, where it could be safely flown. After the drone took off, the patients operated the drone with only their eyes in the order of rotating right, rotating left, ascending, descending, and moving forward. The patients then rotated the drone further to the right and saw Tokai University’s student staff holding flowers and waving their hands, which completed the test.

In the outdoor test, the drone was flown from the rooftop of a building with a good view in Tokai University. After the drone was airborne and the patients saw a student with the message “Welcome to Tokai University,” the patients rotated the drone to the right to enjoy the campus scenery and view images of students who were juggling. There were also students holding a billboard with the message “Thank you for participating.” By rotating further to the right, the patients could see students waving and either a Mt. Fuji photograph or the real scenery. The test was terminated after the patients saw Mt. Fuji (actual or photographed).

#### 2.4.3. Post-Experimental Phase

In this phase, patients who completed the experiment were interviewed regarding their impressions, feelings, difficulties, and overall experience from the start of the experiment to the completion of the study. Significant descriptive terms were noted, particularly those for improving the procedures and robotic systems, as well as patient care in the future. However, because the focus of this study was the description of the role of nurses as intermediaries, patients’ data collected during the post-experimental phase was not included in this study.

### 2.5. Procedure for Data Analysis

The data collected were observation notes of the activities of the intermediaries in the experiment process. The observer (a researcher from the research team) noted significant activities of the events, especially during the intermediaries’ active engagements. The observation data were transcribed and analyzed by noting significant words and phrases highlighting the roles and activities of the intermediaries. These data were grouped together based on the activities and events observed during the pre-experimental phase (calibration process) and the actual experiment process. The recorded videos of the experiment were reviewed to confirm the noted observations.

### 2.6. Ethical Considerations

Ethical approval was obtained from the Ethics Committee of Tokai University (approval number: 21029) and the Mifune Hospital Clinical Research Ethics Review Committee (approval number: 20200408). Before data collection, researchers provided an explanation regarding the study and its procedures to the participants. Written consent was obtained from each participant.

## 3. Results

### 3.1. Patients’ Description

There were 17 patients who participated in the indoor experiment, 8 men and 9 women, with an average age of 69.1 years (standard deviation: 8.25 years). Sixteen of these participants performed the experiment with their naked eyes, and one wore corrective glasses. Twenty-three patients who participated in the outdoor experiment met the selection criteria. There were 11 men and 12 women, with an average age of 68.5 years (standard deviation: 9.00 years). Twenty of them performed the eye-tracking activity with their naked eyes, and three of them wore corrective glasses. Additionally, 15 of the 23 patients also participated in the indoor experimentation. The summary of the indoor and outdoor experiments was presented in Table 1. 

### 3.2. Intermediary Roles

Intermediary roles in this study refer to the actions and roles of nurses that were observed throughout the two phases as described below:

#### 3.2.1. Pre-Experimental Phase

This study defined the “pre-experimental phase” as the setting phase for patients to operate the drone. During this phase, the intermediaries provided support so that the patients could conduct the calibration for eye tracking and be motivated to conduct the calibration. The three roles observed as intermediaries during this phase were:(1)**Advocate**. As an advocate, the intermediaries explained the study procedures to the patients before data were collected, when patients answered questions regarding the study, and they accompanied the patients during the pre-experiment and during communication with the remote operator.(2)**Supporter**. As part of their role as persons who support calibration for eye tracking, the intermediaries provided instructions to the patients based on the instructions of the remote operator and facilitated the calibration process by holding the patient’s head when necessary. To assure a successful calibration process, it is important to keep the head straight while the gaze is moving. However, it was observed that some patients had difficulty moving the gaze without moving the head. When this situation occurred, the intermediaries held the patient’s head with the patient’s permission to avoid unnecessary head movements.

Figure 5 exhibits the picture of the situation from which the following observation notes were derived that depicted the intermediary roles during the pre-experimental phase.
*“Before the experiment start, when the explanation of experiment procedures by video provided for patients, the nurses stayed with the patient and explained to the patient”*.(Observation note 1)
*“The patient moved his head while staring at the dots (the calibration phase), which made the dots did not disappear. Then, the nurse asked for permission to touch the head and hold the head to prevent unnecessary head movement”*.(Observation note 2)

(3)**Motivator.** As a motivator, the intermediaries encouraged the patients to complete the calibration process, particularly when the patients found it difficult to focus the eyes on the dots in the screen. In addition, the intermediaries also praised the patients when they successfully completed the calibration process.

#### 3.2.2. Experimental Phase

During the experimental phase, the intermediaries supported the patients so that they could operate the drone safely. In addition, the intermediaries accompanied the patients during the drone operation. The intermediaries helped the patients to see the scenery of Mt. Fuji live, and/or a photograph of Mt. Fuji in the outdoor experiment. At the end of the experiments, most of the patients reported their experience as “*fun*” and “*I want to fly the drone again*.”

During the Experimental phase, there were three roles observed as intermediaries.
(1)**Facilitator**. During the experiment, the intermediaries communicated the instructions for operation from the remote operator to the patients while also observing the patients. Below is one observation note that represents the intermediary roles during the experimental phase.
*“In the experiment phase, the nurse arranged an appropriate position for the patient in front of the monitor screen and assisted patients to follow the instructions from the operator when moving the drone”*.(Observation 3)(2)**Empathizer.** Additionally, the intermediaries provided supplementary explanations to the patients with hearing loss, repeated explanations to the patients who did not understand the explanation, and shared joy when the drone flew or when the patients succeeded in operating the drone.
*“The nurse clapped the hand and said “Good good” when one patient was successfully moved the drone”*.(Observation 4)(3)**Evaluator.** In cooperation with the ward staff, the intermediaries decided whether to continue the experiment, observed the patient’s mental state and fatigue level when the calibration did not go well, and decided to discontinue when needed.

## 4. Discussion

This article discusses the role of intermediaries in the use of a remote-control drone system used by patients, particularly older patients in psychiatric long-term care. Results showed that the use of the drone controlled by patients was effective in bringing joy to patients, as they could connect with outside scenery both live and in real time. Self-report evaluations from patients, which described their experiences as “fun” and “wanting to fly the drone again”, conveyed their joyful experiences in controlling the drone. Thus, the researchers concluded that the remote-control drone system using eye tracking is effective to bring joy and to help improve the well-being of older people in psychiatric long-term care. These results support the assumption that flying the drone by remote control, using the system with only the eyes, could improve QOL in older patients with mental conditions.

Nowadays, nursing is fast becoming a relationship between persons and intelligent machines. Nurses use technologies of care in their practice [17]. The successful control of the drone by the patients was supported with the critical roles of the intermediaries. The role of intermediaries in transactional relationships between patients and intelligent machines has been discussed by Osaka [12]. The description of the role of intermediaries in such transactional relationships is important in guiding nursing practice in transactive engagements involving intelligent machines [12].

In this study, the patients who participated were older people with mental conditions undergoing long-term care, and some were bedridden; therefore, following the operator’s instructions was not easy for them. Here, the intermediaries played important roles throughout the pre-experimental phase and experiment process.

### 4.1. Pre-Experimental Phase

There were three roles as intermediaries in the pre-experimental phase: **(1) Advocate** for the patient, **(2) Supporter** of the calibration for eye tracking, and **(3) Motivator.**

Being an advocate for the patient is an inseparable part of the role of a nurse. In this study, as the advocate, the intermediaries accompanied the patients through the process, starting from the explanation of the study to the end of the experiment. The intermediaries paid attention to the patients’ needs and connected the patients with the remote operator and the system. Davoodvand et al. [18] explained advocacy in two actions, namely: being in empathy with the patient and protecting the patient. Being in empathy includes understanding, feeling close with the patient, and being sympathetic with the patient, whereas the act of protecting the patient includes the protection of their rights, the completion of the care process, and prioritizing the patient’s health.

Furthermore, the intermediaries, in their role as persons who supported the calibration for eye tracking, assisted the remote operator in the calibration process, gave the instructions to the patients in completing the calibration, and helped to maintain the correct position of their heads until completing the calibration. The intermediaries’ role, as persons who supported the calibration for eye tracking, required the intermediaries’ competency and understanding of the system. This role reflects the technological competency needed by nurses in this technologically demanding care unit. Locsin [19], in his *Technological Competency as Caring in Nursing*, described that technological competency is the expression of caring. In a situation in which technologies are used for patient care, being technologically competent is being caring [19].

There are some difficulties for people with mental illness in psychiatric long-term care in operating a new, advanced technology, i.e., a drone. Therefore, using the motivator as an intermediary, it was thought that it was important to motivate patients, so that they could meet new people and feel they were having fun by practicing and maneuvering the drone.

### 4.2. Experimental Phase

In the experimental phase, the intermediaries played the roles as: **(1) Facilitator, (2) Empathizer** to the patients, and **(3) Evaluator** of patient’s mental state and fatigue level. Overall, the roles of the intermediaries included the roles of advocate, persons who supported calibration for eye tracking, and the facilitators.

As a **facilitator**, the intermediaries aided in the success of the whole process. When the video of the research explanation was shown to the patients, the intermediaries explained the video to the patients. During the experiment, when Mt. Fuji was seen, the intermediaries also explained the view of Mt. Fuji to the patients. The role of the facilitator highlights the intermediaries’ capability of professional communication, particularly in connecting the patient with the remote operator. Ghiyasvandian et al. [20] described nurses’ roles as a facilitator in professional communication as the mediator of communication between patients and other healthcare providers and the executor of others’ duties related to patients’ care.

The **Empathizer** to the patients helped as patients worked on new things, met new people through drones, and shared new experiences in places they had never seen before. In addition, an empathic understanding of sharing the emotions and joys of the patients was an important part of the study. Empathy is an important component of the nurse–patient relationship and nursing care [21]. Successful nursing care can be achieved as nurses bring empathy to their patients.

The **evaluator** of the patient’s mental state and fatigue level was the most important role. The role of nurses and medical staff as intermediaries in assessing the mental state and malaise of persons with mental illness and persons with disabilities was imperative. Evaluating patients’ physiological and psychological conditions are integral parts in the successful evaluation of patients [22].

Furthermore, it was not easy for some older patients to focus and move their eyes in a controlled manner for a long time. Therefore, the drone operation by patients was conducted in a short time of around 5–7 min. In addition, for some older patients who found it difficult to control their eyes and head, the intermediary stayed with these patients during the experiment to help them concentrate and to hold their heads, with their permission, to prevent unnecessary head movement.

### 4.3. Implications for Practice

The future value and relevance of nursing and healthcare are enhanced through interventions associating the outside world among patients, involving advanced and sophisticated technologies, and human caring activities. In this study, this association was fostered by a remotely controlled drone system through which patients with limited access to the outside world, because of disease and other health care conditions such as mental health problems, were provided access and a source of enjoyment. The value of the environment as a metaparadigm of nursing [23] is well-established, and nursing activities that foster human–environment and care–healing conditions are clearly desired. The ability of patients to “see the world” outside of the hospital setting can allow them to live their lives more meaningfully as human beings. By providing this opportunity to patients, nurses render a critical and valuable service as intermediaries in high-tech interventions such as those required in a remotely controlled drone system.

However, the future of high-tech healthcare also requires proficiency with technologies in healthcare. Nurses, as intermediaries, need to be adept with technological know-how, and they must be able to “troubleshoot” technology-based problems with sophisticated technologies in the healthcare arena. It behooves nurses to be technologically prepared to meet the future demand of nurse roboticists, informaticians, and engineers.

## 5. Conclusions

This study found that successfully flying a drone could be competently performed using the eye movements of patients in psychiatric long-term care. As a result, it was determined that, because the patients in psychiatric long-term care were able to view the outside environment of the hospital through the drone system that they controlled, the activity brought them joy, by relating to the outside scenery in real-time and with live action. However, the successful operation of the remotely controlled drone system was mainly supported by nurse intermediaries in their roles as advocates, motivators, and mediators.

## Figures and Tables

**Figure 1 healthcare-10-00827-f001:**
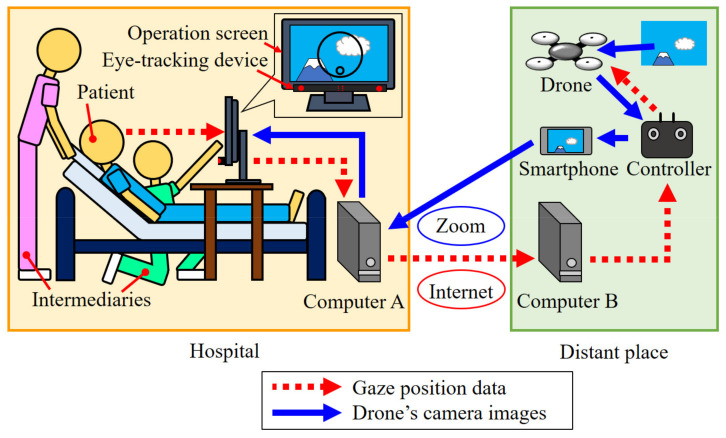
Remote-controlled drone system.

**Figure 2 healthcare-10-00827-f002:**
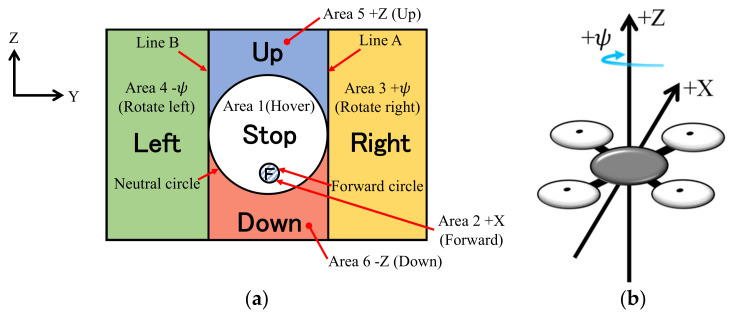
Operation screen and the coordinates around the drone. (**a**) Operation screen and (**b**) Coordinate system.

**Figure 3 healthcare-10-00827-f003:**
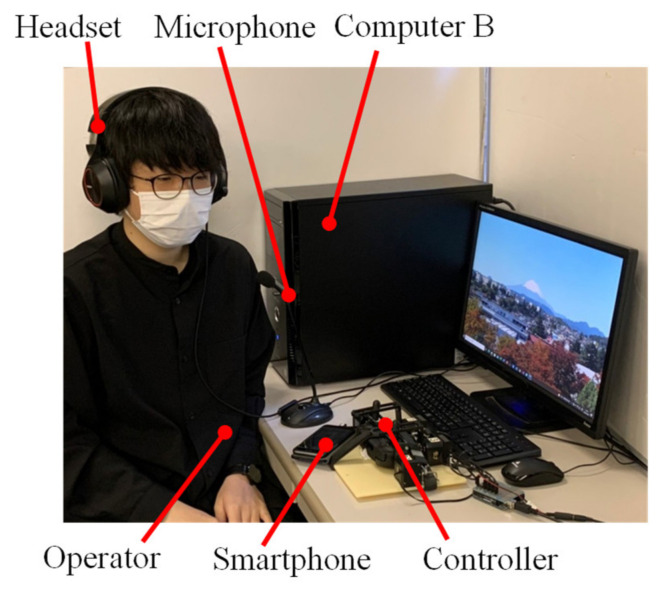
Operator (in Tokai University).

**Figure 4 healthcare-10-00827-f004:**
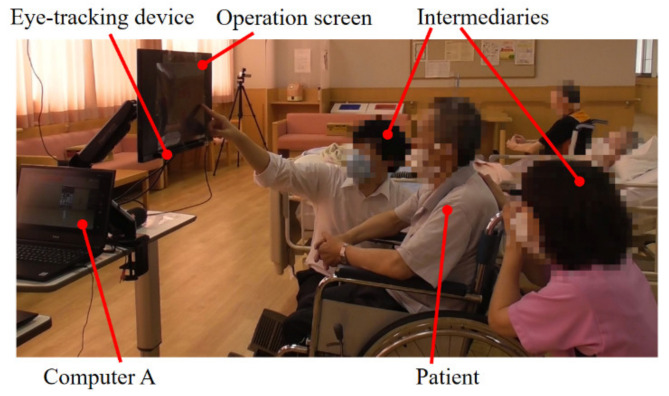
Patient (in the hospital setting).

**Figure 5 healthcare-10-00827-f005:**
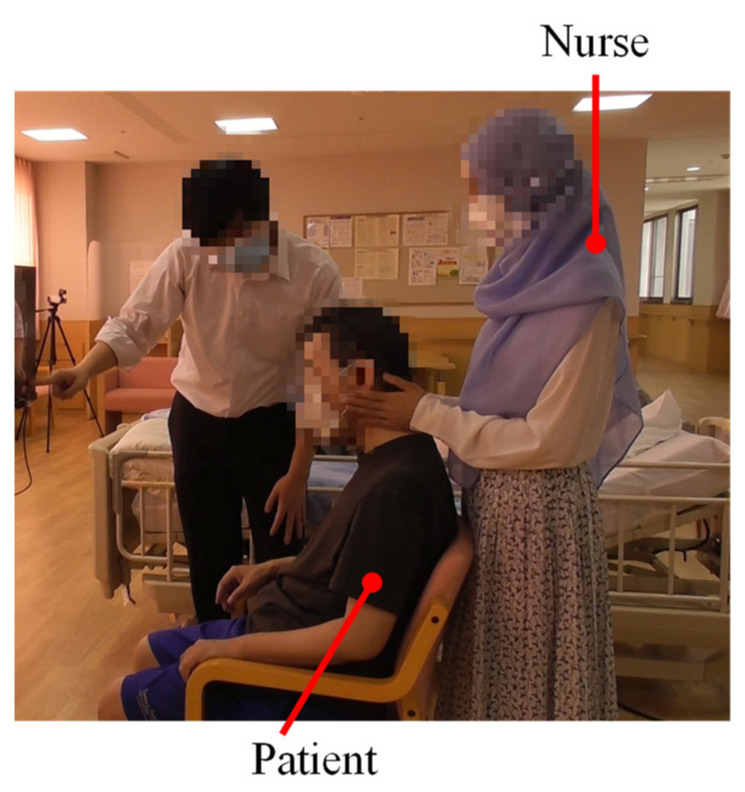
The nurse with the patient.

**Table 1 healthcare-10-00827-t001:** The summary of the indoor and outdoor experiments of the participants.

Descriptions	Indoor Experiment (*N* = 17)	Outdoor Experiment (*N* = 23)
Mean Age (years ± SD)	69.1 ± 8.25	68.5 ± 9.00
Gender		
Male	8	11
Female	9	12
Conditions when moving the drone		
With naked eyes	16	20
With a corrective glass	1	3
Patients’ impression of the experiment		
Enjoy/fun	13	19
Not enjoy	0	0
Neutral	4	4

Note: *N* = Number of patients, SD = standard deviation.

## Data Availability

Data are available on request due to ethical restrictions.

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
