# Peer review of "Remote-Controlled Drone System through Eye Movements of Patients Who Need Long-Term Care: An Intermediary’s Role"

_healthcare, 2022, doi:10.3390/healthcare10050827_

Round 1
Reviewer 1 Report
This work presents a very interesting proposal as well as the obtained information. However, the reviewer strongly suggest that the author properly address the following comments:
1.-Grammatical revision in the use of verbs in the summary and conclusions sections.
2.- Provide tangible and/or quantitative data of the relevant results obtained in the summary and conclusions sections.
3.- In the figure 2, Do not mix legends or representations (symbols) of two languages, only use the English language.
4.- Make the table or graph of the tests of each person when they were satisfactory and unsatisfactory, and what problems they presented, in addition to the positive and negative exclamations.
5.- Why didn't they test more people from other hospitals, because some have more comfort than others?. This is observed in the images, fig. 4
6.- Why in a psychiatric hospital and not in others where there are older people who have other illnesses?
7.- The data that they present from the results obtained, turn them into graphs and tables.
8.-Mention what problems they presented in their communication system, how they developed the algorithms, they used commercial software, if they made applications in commercial operating systems, if they measured data regarding communication times, link times, eye tracking problems, if it was automatic or you had control systems, etc.
9.-This proposal, should meet any medical standard, which or which were?
Reviewer 2 Report
Row 48-55 set the stage for a study of suicidal thoughts and depression. While potentially relevant for this group of users, the emphasis on the mental health aspects does not match what the paper actually reports on in this paper.
Row 93-94 shifts the focus to the intermediary's roles to enable patients eye-tracking capability. This is in line with the title of the paper.
Section 2.2.1 does not explain why there is a need for two separate computers and the link between the controller and a smartphone. I understand there are technical reasons for this (and can guess at several of them), but explaining them would be expected - even if kept on a fairly abstract level given that the paper is not focusing on the technical implementation as contribution. Additionally, while the patient is included in Figure 1, the intermediary who is the focus of this paper is nowhere to be seen in the system overview. Given that the intermediary is your focus, I would expect a system overview where that role was the central element of the figure.
Section 2.2.2 would benefit from clarifying that the operation screen explains where the patient would look in order to control the drone. It is implied, but would be an easy clarification to make in sentence 1 starting on row 122.
Section 2.3, starting on row 145, struggles with the lack of clarification as to where the intermediary comes into the system. The inclusion of nursing staff as intermediaries, operators of the drone, and then some students as well without a specified role, makes the human agents' perspectives hard to get a clear understanding of. Are these all intermediaries or just the nurses? Who are the operators of the drone? What role did the students have and why were they needed? How were they and the nurses selected? Are there specific needs from them?
Section 2.3, starting on row 150, introduces the notion of indoor and outdoor testing which (as far as I have seen) is the first time this is mentioned. I would suggest this part to be moved to the data collection procedure section instead, as it is not yet introduced.
Section 2.4 may benefit from a slightly extended sentence about the content, just clarifying the parts of the experiment phase. A second sentence or just an extended first sentence saying that it is outdoor and indoor experiments would be sufficient.
Section 2.4.1, row 160 and onwards, does not contain a separate calibration process for outdoor and indoor tests. Was that not needed, and if so why not? This section finally starts showing that the operator is actually another human agent and not the patient. I assumed this would be the case earlier, but that was never made clear as per my comments on section 2.2.1. It is also not clear who the operator is - a researcher, a student, or a specially trained drone operator.
Section 2.4.1, row 177, should have been 2.4.2.
Section 2.4.1, row 177 and onwards, does not make it clear to what extent the patient is actually controlling the drone or if the operator is based on instructions from the patient. It is also not clear how the nursing staff has an intermediary role at all. This is a major flaw in the paper as that is the unit of analysis and is given no description in terms of their role, what data is collected from them, and how this data is collected. Only in section 3, row 203, where results are presented does it become clarified that some sort of observation was done of the intermediaries. That is too late, and also lacks detail in terms of how the observations were done and how they were analyzed.
Section 3, contains two main phases: one pre-experimental and one experimental. These should have been described in the method section to address the major flaw I mentioned above (regarding 2.4.1, row 177 and onwards). I am also surprised that there is no post-experimental phase. Was there no follow-up done after the experiment had been concluded? The research does not appear concluded without allowing all human agents - and in particular the intermediaries that are the focus of the paper - reflect on the experiences and hear their thoughts post-experiment, including the nurses perceived impact it has had on the patients.
Section 3.1 lacks in systematic reporting of actual data. The section tells the story of what was done with very limited detail as to differences between intermediaries and/or patients. The section also lacks any kind of reflection as to the pre-experimental expectations of intermediaries and/or patients. Subsequently, the section reads more like a description of what was done rather than an actual results section with tangible data points and summaries of pre-experiment insight. This is obviously a major flaw as the paper does not contain any data related to the phase that could be used to support the discussion.
Section 3.2 similarly does not report any actual data, and is also a description of the phase rather than a results section. Again, this is a major flaw as the paper does not contain any data related to the phase that could be used to support the discussion.
Section 4 moves on to provide anecdotal arguments for suitability and potential, but has no data to support these arguments on. The links to previous research and potential for the technology may well work - once the authors actually present their results and compare them with the results of the references they lean on. Until that is addressed, the paper could have been written without any actual study being run as no data at all is presented.
Reviewer 3 Report
An Intermediary’s Role When Using a Remote-Controlled 2 Drone System by Means of Eye Movements of Patients in 3 Long-term Care
Reviewer’s comments
This is an original article with novelty. The subject matter would be of interest to readers. A general suggestion is that if more attention is paid to construction of sentences the length of the article could be reduced . Brevity helps the reader to comprehend the take home message easily and quickly. For example the title could be re written as Remote controlling of Drones through eye movements of geriatric institutionalised patients. An Intermediary’s Role . Surprisingly there are no references dealing with use of eye movements in operating drones eg
- ‘Human Gaze-Driven Spatial Tasking of an Autonomous MAV’, written by Liangzhe Yuan, Christopher Reardon, Garrett Warnell, and Giuseppe Loianno, from the University of Pennsylvania, U.S. Army Research Laboratory, and New York University.
- https://dronelife.com/2018/09/26/drone-eye-tracking-pilot/
- GazeGuide: An Eye-Gaze-Guided Active Immersive UAV Camera. Pavan Kumar B. N.,Adithya Balasubramanyam,,Ashok Kumar Patil, Chethana B. and Young Ho Chai -Appl. Sci. 2020, 10(5), 1668; https://doi.org/10.3390/app10051668
The authors pre suppose that all readers are familiar with how exactly eye tracking data is converted to lifting a drone. This needs to be elaborated for the better understanding of the non expert.
Round 2
Reviewer 1 Report
This paper has been improved, thank you for addressing my comments.
